# Towards High-resolution 3D Anomaly Detection via Group-Level Feature Contrastive Learning

## ABSTRACT

High-resolution point clouds (HRPCD) anomaly detection (AD) plays a critical role in precision machining and high-end equipment manufacturing. Despite considerable 3D-AD methods that have been proposed recently, they still cannot meet the requirements of the HRPCD-AD task. There are several challenges: i) It is difficult to directly capture HRPCD information due to large amounts of points at the sample level; ii) The advanced transformer-based methods usually obtain anisotropic features, leading to degradation of the representation; iii) The proportion of abnormal areas is very small, which makes it difficult to characterize. To address these challenges, we propose a novel group-level feature-based network, called Group3AD, which has a significantly efficient representation ability. First, we design an Intercluster Uniformity Network (IUN) to present the mapping of different groups in the feature space as several clusters, and obtain a more uniform distribution between clusters representing different parts of the point clouds in the feature space. Then, an Intracluster Alignment Network (IAN) is designed to encourage groups within the cluster to be distributed tightly in the feature space. In addition, we propose an Adaptive Group-Center Selection (AGCS) based on geometric information to improve the pixel density of potential anomalous regions during inference. The experimental results verify the effectiveness of our proposed Group3AD, which surpasses Reg3D-AD by the margin of 5% in terms of object-level AUROC on Real3D-AD.

## CCS CONCEPTS

• **Computing methodologies** → **Visual inspection**; **Anomaly detection**.

## KEYWORDS

Anomaly Detection, 3D Point Clouds, Contrastive Learning, Feature Representation

## 1 INTRODUCTION

Anomaly Detection (AD) is a critical field in machine learning aimed at identifying unusual patterns or abnormalities that do not conform to expected behavior. Traditionally, AD has been extensively applied in 2D image analysis, where methods primarily focus on identifying anomalies through pixel-level discrepancies [25, 39]. However, these 2D techniques come with inherent limitations, such

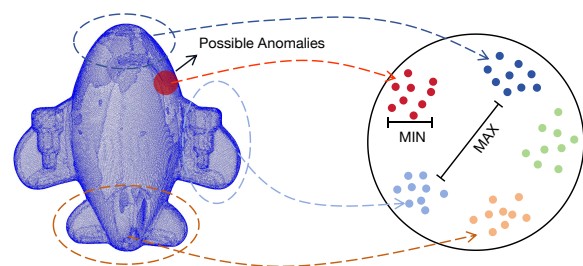

**Figure 1: Ideal feature distribution of normal point clouds and abnormalies for the high-resolution 3D anomaly detection task. Group-level features are used to express structural information.**

as the fixed perspectives and inability to capture complex geometries, which often result in loss of important spatial information and context [3, 8, 10, 22, 25, 39, 40]. Transitioning from 2D images to 3D Point Clouds (PCD) effectively overcomes these limitations. 3D PCDs, which are sets of data points in space, provide a more comprehensive representation of objects by capturing their shape and spatial hierarchy in real environments [4, 16].

Despite the advantages, the field of 3D-AD faces many challenges [9, 24]. The High-Resolution (HR) of 3D PCDs introduces computational and analytical complexities due to the sheer increase in data volume and the intricacies involved in 3D space analysis [21, 24]. Research based on HRPCD-AD has just emerged, and how to construct efficient representations for AD tasks in HRPCD has become a major challenge. There is an urgent need to improve the precision of HRPCD-AD to meet the needs of industrial manufacturing (IM) [21, 24]. The obstacle on the road to establishing efficient representation is threefold. i) Existing HRPCD networks necessitate downsampling for large datasets, risking the loss of crucial anomaly information. ii) Recent 3D representation methods yield embeddings with insufficient distinction in AD. iii) The proportion of anomalies in HRPCD is small and obscure, making it easy to overlook [18, 21, 24, 25, 39].

To make the training of the network free from the constraints of HR PCDs and too few training samples, we introduce the idea of group-level feature enhancement. Existing HRPCD networks clearly require downsampling operations when faced with items composed of millions of points during training or inference [21, 24]. Among them, although Reg3D-AD [24] has a higher group density in HRPCD reasoning in a way similar to PatchCore [29], it still cannot be directly trained by HRPCD. There is a gap in the middle, which affects the performance. IMRNet also realized the drawbacks of downsampling and proposed adaptive sampling, but still did not fundamentally solve the problem. We need to study a solution that is not constrained by point cloud resolution. Unlike previous 3D contrastive learning studies [1, 23, 26], we build contrastive

learning at group level in a single sample to makes our network training unconstrained by point cloud resolution and scale. The divide and conquer design makes our network's group batchsize infinitely large in theory, although its batchsize is 1.

To achieve better representation, we pose a key question: **how to create an ideal distribution required by HRPCD-AD in the feature space?** Currently, most network backbones that perform well on HRPCD use the Point Transformer architecture [21, 24]. The embeddings represented by this encoder exhibit anisotropic distribution in the feature space. Research [12, 13, 37] has shown that transformer-based structures place more emphasis on local contextual information in representation, while ignoring global semantic information. This results in high-frequency groups being distributed in a narrow area and close to the center in the feature space. While low-frequency groups are relatively sparse and far from the origin. Commonly, calculating feature similarity is a widely-used method, but this anisotropic spatial distribution will bring problems to the measurement of feature similarity. Figure 1 shows an ideal feature space distribution. Specifically, excellent spatial distribution of features should have both uniformity and alignment [7, 14]. Uniformity requires that the vectors should be distributed as widely as possible and isotropic in the space, while alignment means that the distance between similar vectors in the feature space should be small. Meanwhile, the excellent encoder should be capable of higher anomaly sensitivity to easily distinguish abnormal representation from normal representation in the feature space. Considering above-mentioned distribution characteristics: **intercluster dispersion** and **intracluster compactness**, we propose to construct the group-level contrastive learning architecture. The proposed Intercluster Uniformity Network (IUN) pushes clusters representing similar features far away in the feature space to enhance the uniformity of feature distribution. On the basis of IUN, the Intracluster Alignment Network (IAN) further tightens the features in the same cluster to strengthen the alignment of feature distribution.

To better capture subtle features, we design the Adaptive Group-Center Selection (AGCS). Drawing inspiration from the practices of quality inspection engineers, who often focus more intensively on areas they suspect to be problematic to enhance the detection of abnormalities, AGCS similarly prioritizes regions within 3D point clouds that exhibit potential anomalies. Leveraging the Fast Point Feature Histogram (FPFH) [32], AGCS intensifies the examination of areas with significant local geometric variations. Finally, AGCS enables our network to preserve more potential anomaly groups during inference, thereby enhancing its AD capability.

Our contributions are succinctly outlined as follows:

- A novel Group3AD framework for the HRPCD-AD task is designed, which optimizes the spatial information of 3D point clouds, enhancing the precision of anomaly detection.
- We propose the Intercluster Uniformity Network (IUN) and the Intracluster Alignment Network (IAN), which can disperse features across clusters and tighten features within clusters in the feature space, respectively. Both two networks enhance feature uniformity and alignment, improving the coherence of feature representations for anomaly identification and detection accuracy.

- An efficient Adaptive Group-Center Selection (AGCS) is designed, which focuses on regions with potential anomalies, enhancing model sensitivity and detection efficiency.
- The proposed Group3AD is flexible and scalable. Group3AD can be directly integrated with other network architectures, which promotes the wider application of various anomaly detection tasks.

## 2 RELATED WORK

### 2.1 2D Anomaly Detection

Since the introduction of the MVTec AD dataset [2], 2D image anomaly detection (2D-AD) has garnered increased focus [25, 39]. Predominantly, research in this domain utilizes this 2D dataset for exploring unsupervised AD techniques [3, 10, 11, 15, 15, 20, 27, 29, 30, 30, 34, 36, 41, 42]. The intention of image reconstruction-based 2D-AD methods is to reconstruct abnormal images into approximate normal images and achieve anomaly localization through pixel-level comparison. The feature extraction based 2D-AD methods strives to provide more informative embeddings, with more significant differences between normal and anomalous features. Due to the fact that many networks in the former use the method of training from scratch, their performance may be inferior to that of the latter, which uses robust pre-training models.

Notable examples include DRAEM [41], SimpleNet [27], etc. They create fake samples and identify anomalies through reconstruction comparison, building supervised tasks in unsupervised datasets. The representation of pre-trained networks has been proven to be more powerful and effective. PatchCore [29] employs a memory-efficient representation of normal data distribution through sparse sampling of feature space patches for identifying outliers. Cut-Paste [20] markedly enhances AD capabilities by artificially generating anomalies through cutting and pasting segments within images. Self-Taught AD [3] leverages a student-teacher framework where features from a pre-trained teacher network guide a student network trained on normal data to detect anomalies based on feature discrepancies. STPM [36] and MKD [33] both employ a teacher-student architecture for AD, with STPM harnessing multi-scale features directly, while MKD concentrates on distilling knowledge from these multi-scale features through a more efficient network architecture to enhance performance. Normalizing flow [19] methods have created a feature distribution with anomaly sample deviation, expanding its potential. Although 2D-AD methods cannot be directly used for 3D-AD, they highlight the importance of building an easily distinguishable feature distribution and enhancing the representational ability of the network [25, 39].

### 2.2 RGB-D Anomaly Detection

Despite significant advancements in 2D image anomaly detection, the exploration of anomaly detection in 3D PCDs remains relatively underdeveloped [5, 6, 9]. This field saw a notable surge in interest following the release of the pioneering real-world 3D anomaly detection dataset, known as MVTec 3D-AD [4]. This dataset has sparked new investigations into the complexities and potentialities of detecting anomalies within 3D environments. The key challenge in 3D-AD lies in the effective harnessing of depth information, which can significantly enhance detection capabilities in certain

scenarios. This complexity, alongside the potential for improved accuracy, positions 3D-AD as both a challenging and promising avenue for future research.

While AST [31] achieves good results on the MVTec 3D-AD by utilizing depth information to separate the background, it primarily depends on 2D-AD techniques for anomaly detection, neglecting the depth characteristics of objects. Designing the feature extraction module for detecting anomalies in PCD demands fresh approaches. Recent studies have made efforts to design networks with stronger point clouds representation capabilities [8]. 3D-ST [5] introduces a self-supervised learning strategy for representation learning in PCD and employs a knowledge distillation-based modeling module. While CPMF [8] streamlines anomaly detection by projecting point clouds into 2D images from various angles to reduce feature extraction complexity and computational costs, this approach does not fully leverage the advantages inherent in the 3D nature of point clouds data. BTF [17] underscores the efficacy of traditional hand-crafted PCD descriptors, yet notes the underperformance of learned features. BTF attributes this paradox to the inadequate transferability of existing pretrained features on the current small-scale PCD datasets. The use of spatial information for AD tasks still deserves further exploration [8, 21, 24, 38].

### 2.3 High-resolution 3D Anomaly Detection

Some methods have made efforts in the direction of PCD anomaly representation. However, MVTec 3D-AD [4] is an RGB-D dataset with low resolution, which cannot further explore the value of spatial information in AD tasks. The proposal of the Real3D-AD [24] dataset containing HR multi-view information of objects brings more development space to AD. The point resolution and precision of Real3D-AD are 4.28 and 9 times higher than MVTec 3D-AD, respectively. The ultra-high accuracy brings more potential and hope to AD tasks but poses significant challenges to establishing representations. CPMF [8] aims to achieve PCD AD by merging handcrafted PCD descriptions with the capabilities of pre-trained 2D neural networks. CPMF performs well on the RGBD dataset, but poorly on Real3D-AD. Reg3D-AD [24] is a benchmark method built on Real3D-AD, which is built on the memory bank and achieves significant performance improvements using pre-trained models. However, the pre-training of Reg3D-AD is difficult to establish directly on HRPCD datasets. IMRNet [21] introduces a self-supervised, scalable framework for 3D PCD AD, leveraging iterative mask reconstruction and geometry-aware sampling to identify and localize anomalies with high accuracy. However, IMRNet has also been affected by the negative performance impact of downsampling.

In view of this, our work aims to design a method that fully utilizes the HRPCD spatial information to enhance the model's generalization ability with small samples in unsupervised manner.

## 3 APPROACH

### 3.1 Problem Definition

Our approach to 3D-AD aligns with the settings defined by Real3D-AD [24], emphasizing the intricacies of handling HRPCD. Defined formally, our task involves a set of training examples $T = \{t_i\}_{i=1}^{N}$, where each $t_i$ is a normal point clouds sample, belonging to a specific category $c_j$, with $c_j \in C$ where $C$ represents the entire set

of categories. In testing, determine whether a sample in a certain category contains anomalies, and if so, define the entire sample as an anomaly and locate the anomaly area. The HR 3D-AD dataset is unique in that it exclusively comprises point clouds, offering a more nuanced and comprehensive representation of objects compared to traditional RGB-D (treated as 2.5D) datasets, which may not meet the precision requirements of industrial manufacturing due to potential blind spots from single-view scanning.

### 3.2 Group3AD

This section delves into the detailed architecture of Group3AD, as shown in Figure 2, a model specifically designed to enhance the resolution and accuracy of 3D anomaly detection through group-level feature contrastive learning. Group3AD is composed of three principal components: Intercluster Uniformity Network (IUN), Intracluster Alignment Network (IAN), and Adaptive Group-Center Selection (AGCS). These modules work in synergy to optimize feature representation, thereby improving the precision and efficiency of anomaly detection.

*3.2.1 Intercluster Uniformity Network (IUN).* The IUN establishes an intercluster contrastive learning task. Each cluster is composed of group-level feature vectors with similarity. This network sets each cluster as negative samples in contrastive learning, and widens the distance between each cluster to obtain a uniform structured sample space.

We construct fake anomalies on the original HR point clouds so that the encoder can still maintain uniformity when encoding anomaly groups during inference. Given the original point clouds $P$, $P_i$ represents the $i$th point in $P$. Firstly, calculate the FPFH features of the point clouds:

$$FPFH(P_i) = ComputeFPFH(P_i, \text{neighbors}(P_i)). \quad (1)$$

Next, select the point $c$ with the highest FPFH feature as the center of the local region:

$$c = \text{argmax}_{p_i} FPFH(P_i). \quad (2)$$

Then, based on the center point $c$, a random $r \in [1\%, 10\%]$ of the total point clouds will be selected around c as the local area:

$$P_{local} = \{P_j \mid \text{distance}(P_j, c) \leq d, j = 1, ..., N\}, \quad (3)$$

where $d$ is a distance threshold that ensures $P_{local}$ contains approximately $r$ of the points in $P$ are included. Subsequently, randomly select a value with a variance $\sigma^2 \in [0.01, 0.05]$ and use $\sigma^2$ to generate a normally distributed random noise $\epsilon$ with an average value of 0. Add noise to each dimension of $P$ to generate anomaly regions $P_{anomaly}$:

$$P_{\text{anomaly},i} = P_{\text{local},i} + \epsilon, \quad (4)$$

where $\epsilon = \mathcal{N}(0, \sigma^2)$. Repeat this process for all $P$ midpoints to obtain the set of anomalies $A$:

$$A = P_{\text{anomaly},i} \mid \forall P_{\text{local},i} \in P_{\text{local}}. \quad (5)$$

Combine $A$ and $P$ to obtain $FakeAnomalyP$:

$$FakeAnomaly(P) = P \cup A. \quad (6)$$

We cluster group-level features using the K-means [28] algorithm to determine which cluster $(P_i)$ belongs to. K-means determines the number of clusters using the Elbow Method [35]. Determine

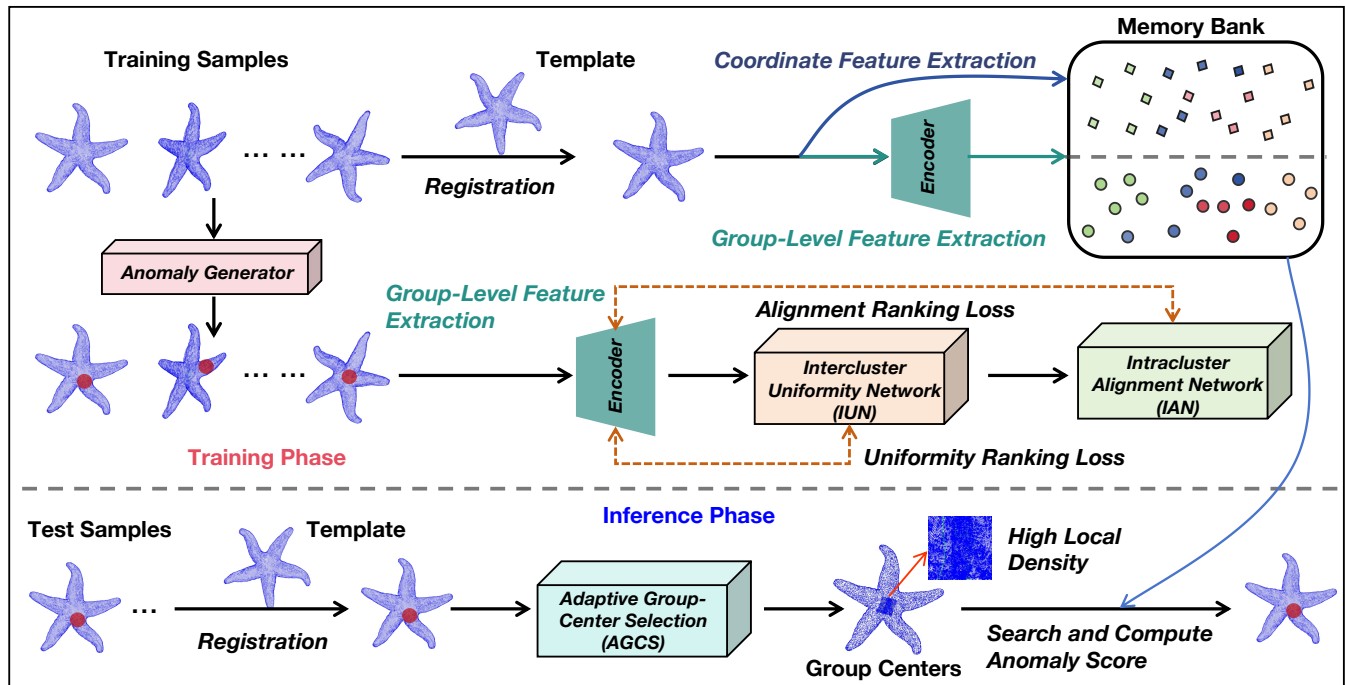

**Figure 2: The pipeline of Group3AD, which consists of three main components. (1) Group-Level Feature Extraction extracts group-level features from the input 3D point clouds. (2) Intercluster Uniformity Network (IUN) and Intracluster Alignment Network (IAN) enhance the feature separation between clusters and tighten the distribution within clusters, respectively, for improving anomaly detection accuracy. (3) Adaptive Group-Center Selection (AGCS), used during inference, dynamically focuses on regions with potential anomalies by adjusting the sampling density based on geometric information. This structured approach ensures efficient anomaly detection in complex 3D environments.**

the final size of $K$ by plotting the relationship between Within Cluster Sum of Squares (WCSS) and the number of clusters ($K$). Find the "Elbow" point in the relationship diagram, which is the position where the rate of WCSS decrease suddenly slows down as $k$ increases. This point is usually considered the optimal number of clusters, as it represents the best balance between the intracluster compactness and the number of clusters.

The calculation of WCSS is represented as:

$$W_k = \sum_{x_i \in C_k} \|X_i - \mu_k\|_2^2, \tag{7}$$

where $\mu_k$ is the $k$th cluster center, $C_k$ is the point set in the $k$th cluster, $W_k$ is the WCSS of the $k$th cluster, and $\| X_i - \mu_k \|^2$ is the square of the Euclidean distance from point $x_i$ to the cluster center $\mu_k$. The entire dataset $W$ is the sum of all clustered WCSS:

$$W = \sum_{k=1}^{K} W_k. \tag{8}$$

Assuming we have $k$ cluster centers, denoted as $C = \{C_1, C_2, ..., C_k\}$, where each cluster center $C_i$ is an $n$-dimensional vector. For any two different cluster centers $C_i$ and $C_j$, we calculate the Euclidean distance between them:

$$d\left(C_i, C_j\right) = \left\|C_i - C_j\right\|_2, \tag{9}$$

where $\| \cdot \|$ represents calculating Euclidean distance. We compare the distances between each pair of cluster centers and select the minimum value among these distances:

$$\text{min\_dist} = \min_{1 \le i < j \le K} \text{distance}\left(C_i, C_j\right). \tag{10}$$

Define the uniformity ranking loss function as the reciprocal of this minimum distance:

$$L_{uniformity} = \frac{1}{\text{min\_dist}}. \tag{11}$$

The larger the value of the uniformity ranking loss function, the smaller the minimum distance between cluster centers, the lower the discrimination between clusters, and the greater the loss. Therefore, our goal is to minimize this loss function, thereby maximizing the minimum distance between cluster centers and improving the quality of clustering. The changes in the distribution of vectors in the feature space represented by the optimized encoder through the uniformity ranking loss are shown in Figure 3(b).

*3.2.2 Intracluster Alignment Network (IAN).* The IAN establishes an intracluster contrastive learning task. Each cluster constitutes one mini-batch. Each vector in the cluster constitutes one positive sample in contrastive learning. Assuming we have $k$ cluster, denoted as $C = \{C_1, C_2, ..., C_k\}$, where each cluster $C_k$ contains a certain number of points. We use $P_k = \{P_{k1}, P_{k2}, \ldots, P_{kn}\}$ to represent the

(a) Original Feature Distribution

(b) IUN Feature Distribution

(c) IAN Feature Distribution

**Figure 3: Flowchat of group-level feature distribution, constrained by IUN and IAN. The basic idea is to minimize the intra-group distance and maximize the inter-group distance.**

set of points $C_k$ contains. For any two points $P_{ki}$ and $P_{kj}$ in cluster $C_k$, we calculate the Euclidean distance between them:

$$d\left(P_{ki}, P_{kj}\right) = \left\|P_{ki} - P_{kj}\right\|, \tag{12}$$

where $\|\cdot\|$ represents calculating Euclidean distance. For each cluster $C_k$, we need to find the maximum distance between all pairs of internal points:

$$\max\_dist_k = \max_{P_{ki}, P_{kj} \in P_k} d\left(P_{ki}, P_{kj}\right), \tag{13}$$

This means that we compare the distances between all possible point pairs within cluster $C_k$ and select the maximum value among these distances. Finally, we define the alignment ranking loss function as the average of the maximum intracluster distance of all clusters:

$$L_{\text{alignment}} = \frac{T}{k} \sum_{k=1}^{K} \max\_dist_k^2, \tag{14}$$

where $T$ is a temperature coefficient that makes the training of the network more stable. A larger intracluster distance usually means that the points within the cluster are more dispersed. The larger the value of this loss function, the more scattered the points within the cluster, the lower the compactness of the cluster, and the greater the loss. Therefore, our goal is to minimize this loss function and improve the compactness of clustering.

*3.2.3 Adaptive Group-Center Selection (AGCS).* Although our method does not require sampling, we still need to use the Farthest Point Sampling (FPS) algorithm to select the center points of the groups. Obtaining more groups in areas with potential anomalies can enable the model to inject more computing power into these areas to achieve stronger perceptual capabilities. In operation, our goal is to allocate higher groups resolution in areas with significant geometric changes, inspiring attention allocation mechanisms for quality inspection engineers in their work.

To evaluate the geometric characteristics of points, we employ the Fast Point Feature Histogram (FPFH) approach. The FPFH descriptor is utilized to analyze variations in surface geometry by comparing a point with the estimated surface normals of its neighboring points. For each point $p$ within the PCD $P$, We calculate the surface normal of this point, denoted as $n(\cdot)$. We take $n$ neighboring points around point $p$, denoted as $\{N_1, N_2, ..., N_n\}$. We calculate the surface normal difference between point $p$ and its $n$ neighboring points through Euclidean distance, and obtain the surrounding

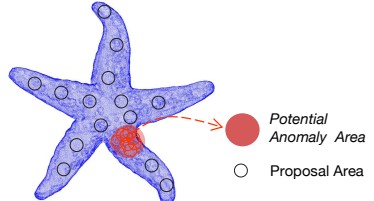

*Potential Anomaly Area*

○ Proposal Area

**Figure 4: Adaptive group-center selection (AGCS). AGCS adaptively selects the points most likely to be within the anomaly region in the selection of group centers via FPFH features.**

variation feature $F$. We have:

$$F_p = \sum_{i=1}^{N} \|f(p) - f(N_i)\|_2. \tag{15}$$

This metric enables us to quantify the dissimilarity in geometric attributes between a point and its neighboring points based on their FPFH descriptors. Such analysis provides crucial insights into the local geometry of the point clouds. In the sampling process of center points, it is assumed that a total of $n$ center points are required. The center points $C_{FPFH}$ sampled by the FPFH method are $a \times n$, and the remaining $(1 - a) \times n$ center points $C_{FPS}$ are obtained through FPS sampling, where $a$ represents the assigned weight. Ultimately, all center points $C$ are composed of both:

$$C = C_{FPFH} \cup C_{FPS}. \tag{16}$$

The AGCS is implemented as detailed in Algorithm 1.

---

**Algorithm 1** Adaptive Group-Center Selection (AGCS)

---

1: **Input:** Point clouds **P**, number of points $N$, attention factor $\alpha$
2: **Output:** Set of center points $\mathbf{C}_{\text{points}}$
3: **function** AGCS(**P**, $N$, $\alpha$)
4:     $N \leftarrow$ ComputeNormals(**P**) ▷ Compute surface normals for each point in the point clouds **P**
5:     $\mathbf{F}_{\text{FPFH}} \leftarrow$ ComputeFPFH(**P**, **N**) ▷ Compute FPFH features for the point clouds based on the surface normals
6:     $\mathbf{F}_{\text{var}} \leftarrow$ ComputeLocalVariation($\mathbf{F}_{\text{FPFH}}$) ▷ Evaluate local geometric variation, indicative of potential anomalies
7:     $\mathbf{P}_{\text{high\_var}} \leftarrow$ SelectHighVariationPoints($\mathbf{F}_{\text{var}}$, $\alpha$) ▷ Select points with high variation in their local geometric features, which are more likely to be near anomalies
8:     $\mathbf{C}_{\text{FPFH}} \leftarrow$ FPS($\mathbf{P}_{\text{high\_var}}$, $N \times \alpha$) ▷ Sample using FPS on high variation points
9:     $\mathbf{C}_{\text{FPS}} \leftarrow$ FPS(**P**, $N \times (1 - \alpha)$) ▷ Sample remaining points uniformly
10:     $\mathbf{C}_{\text{points}} \leftarrow \mathbf{C}_{\text{FPFH}} \cup \mathbf{C}_{\text{FPS}}$
11:     **return** $\mathbf{C}_{\text{points}}$
12: **end function**

---

*3.2.4 Overall Pipeline.* In the Group3AD framework, the **training phase** starts with a pre-trained encoder to extract features from 3D point clouds initially. The Intercluster Uniformity Network (IUN) subsequently enhances the encoder's capabilities, which increases the separation between different clusters to improve the distinctness

of feature representations. Additionally, the Intracluster Alignment Network (IAN) further refines these features by ensuring they are tightly aligned within each cluster. The enhanced encoder then populates a memory bank with these optimized features. During the **inference phase**, the Adaptive Group-Center Selection (AGCS) assists by selectively targeting potential anomaly regions, comparing these regions against the memory bank to accurately compute anomaly scores.

The training and inference implementation of Group3AD are described in Algorithm 2. We denote $\mathbf{P}$ as a set of point clouds from the training or testing dataloader, and use $\mathbf{P}_{mod}$ for the modified point clouds after generating fake anomalies, $\mathbf{F}$ for features extracted by the encoder $\Phi_{enc}$. $C$ and $\mathbf{C}_{centers}$ are for clusters and their centers, respectively. $\mathcal{L}_{IUN}$ and $\mathcal{L}_{IAN}$ are for the loss functions of the IUN and IAN phases. $\mathbf{C}_{points}$ is for the center points selected by AGCS. $\mathbf{S}$ is for the computed anomaly scores during inference.

---

**Algorithm 2** Group3AD with Two-Phase Training

---

1: **Input:** Training dataloader $\mathcal{D}_{train}$, Testing dataloader $\mathcal{D}_{test}$, epochs $E$
2: **Output:** Trained Encoder $\Phi_{enc}$, IUN $\Phi_{IUN}$, IAN $\Phi_{IAN}$, and AGCS settings
3: Initialize $\Phi_{enc}$, $\Phi_{IUN}$, and $\Phi_{IAN}$ randomly
4: /*Phase 1: Training IUN*/
5: **for** $i = 1$ to $E/2$ **do**
6:     **for** Pointclouds $\mathbf{P}$ from $\mathcal{D}_{train}$ **do**
7:         $\mathbf{P}_{mod} \leftarrow$ generate_fake_anomalies($\mathbf{P}$)
8:         $\mathbf{F} \leftarrow \Phi_{enc}(\mathbf{P}_{mod})$
9:         $C, \mathbf{C}_{centers} \leftarrow$ cluster_features($\mathbf{F}$)
10:         $\mathcal{L}_{IUN} \leftarrow$ compute_IUN_loss($\mathbf{C}_{centers}$)
11:         Perform backpropagation on $\mathcal{L}_{IUN}$
12:     **end for**
13: **end for**
14: /*Phase 2: Training IAN*/
15: **for** $i = E/2 + 1$ to $E$ **do**
16:     **for** Pointclouds $\mathbf{P}$ from $\mathcal{D}_{train}$ **do**
17:         $\mathbf{F} \leftarrow \Phi_{enc}(\mathbf{P})$
18:         $C \leftarrow$ cluster_features($\mathbf{F}$) ▷ Assuming the function also returns clusters $C$
19:         $\mathcal{L}_{IAN} \leftarrow$ compute_IAN_loss($C$)
20:         Perform backpropagation on $\mathcal{L}_{IAN}$
21:     **end for**
22: **end for**
23: /*Inference Phase*/
24: **for** Pointclouds $\mathbf{P}$ from $\mathcal{D}_{test}$ **do**
25:     $\mathbf{C}_{points} \leftarrow$ AGCS.select_center_points($\mathbf{P}$)
26:     $\mathbf{F} \leftarrow \Phi_{enc}(\mathbf{P})$
27:     $\mathbf{S} \leftarrow$ compute_anomaly_scores($\mathbf{F}, \mathbf{C}_{points}$) ▷ Further processing based on $\mathbf{S}$
28: **end for**

---

# 4 EXPERIMENTS

## 4.1 Experimental Settings

*4.1.1 Datasets.* In our experiments, we utilize the Real3D-AD [4] dataset. Real3D-AD is the respresentive HR multi-view scanning 3D dataset derived from real-world scenarios. Real3D-AD, comprising 1,254 HR 3D items with each ranging from forty thousand to millions of points, stands as the most extensive dataset for high-precision 3D industrial anomaly detection. Real3D-AD exceeds other available datasets for 3D anomaly detection in terms of point clouds resolution (0.0010mm-0.0015mm), comprehensive 360-degree coverage, and flawless prototype quality.

*4.1.2 Evaluation Metrics.* Our study standardizes the evaluation by employing metrics specifically designed for 3D-AD, which are identical to those detailed in Real3D-AD. We assess the performance of anomaly detection at both the object and point levels using the Area Under the Receiver Operating Characteristic Curve (AUROC) and the Area Under the Precision-Recall Curve (AUPR/AP). Superior anomaly detection capabilities are indicated by higher values of AUROC and AUPR. All tests are performed using a hardware setup comprising a 13th Gen Intel(R) Core(TM) i7-13700K CPU, 32GB DDR5 SDRAM, and a GeForce RTX 3090 graphics card.

## 4.2 Results and Analysis

*4.2.1 Anomaly Detection on Real3D-AD.* Tables 1-4 summarize per-class comparisons between Group3AD and other state-of-the-art methods, namely Reg3D-AD [24], CPMF [8], IMRNet [21], and several benchmarking methods reported in Real3D-AD [24].

(1) **O-AUROC**. Regarding the O-AUROC metric, while Reg3D-AD [24] sets the benchmark among current approaches with an average O-AUROC of 0.704, this mark falls short of being entirely effective. The introduced Group3AD method, however, markedly surpasses these existing standards, registering an impressive O-AUROC of 0.751, illustrating a significant advancement over prior techniques, as shown in Table 1.

(2) **O-AUPR**. In the context of O-AUPR, Reg3D-AD leads among contemporary strategies with an average performance of 0.723, indicating a gap towards optimal precision-recall balance. The newly developed Group3AD method exceeds these precedents, demonstrating a notable O-AUPR of 0.74, showcasing a substantial improvement in accurately identifying anomalies, as shown in Table 2.

(3) **P-AUROC**. Our findings indicate that Group3AD exhibits a marked improvement over Reg3D-AD in the context of P-AUROC scores across various test scenarios. While Reg3D-AD achieved a P-AUROC of 0.705, Group3AD advanced this benchmark to 0.735, indicating a significant enhancement in detection capability, as shown in Table 3. This improvement demonstrates the robustness of Group3AD in navigating the complexities of 3D anomaly detection.

(4) **P-AURR**. Reflecting on the P-AURR metric, we focus on how Group3AD compares to the established benchmark method, Reg3D-AD. Reg3D-AD sets a solid foundation with a P-AURR score of 0.109. Our novel approach, Group3AD, significantly enhances this benchmark by achieving a P-AURR of 0.137, indicating a marked improvement in recall capabilities, as shown in Table 4.

In reviewing the performance across O-AUROC, O-AUPR, P-AUROC, and P-AURR metrics, it is clear that while Reg3D-AD [24] has set solid benchmarks, Group3AD significantly surpasses these, demonstrating exceptional improvements (performance improvements of 6.7%, 2.4%, 4.3%, and 25.7% respectively). This performance solidifies Group3AD's status as a powerful tool for precise and reliable anomaly detection in industrial settings.

**Table 1: O-AUROC score (↑) on Real3D-AD. The best and second-best results are marked in red and blue, respectively.**

| Method | Airplane | Car | Candybar | Chicken | Diamond | Duck | Fish | Gemstone | Seahorse | Shell | Starfish | Toffees | Average |
|---|---|---|---|---|---|---|---|---|---|---|---|---|---|
| BTF(Raw) | 0.520 | 0.560 | 0.462 | 0.432 | 0.545 | 0.784 | 0.549 | 0.648 | 0.779 | 0.754 | 0.575 | 0.630 | 0.603 |
| BTF(FPFH) | 0.730 | 0.647 | 0.703 | 0.789 | 0.707 | 0.691 | 0.602 | 0.686 | 0.596 | 0.396 | 0.530 | 0.539 | 0.635 |
| M3DM(PointBERT) | 0.407 | 0.506 | 0.442 | 0.673 | 0.627 | 0.466 | 0.556 | 0.617 | 0.494 | 0.577 | 0.528 | 0.562 | 0.538 |
| M3DM(PointMAE) | 0.434 | 0.541 | 0.450 | 0.683 | 0.602 | 0.433 | 0.540 | 0.644 | 0.495 | 0.694 | 0.551 | 0.552 | 0.552 |
| PatchCore(FPFH) | 0.882 | 0.590 | 0.565 | 0.837 | 0.574 | 0.546 | 0.675 | 0.370 | 0.505 | 0.589 | 0.441 | 0.541 | 0.593 |
| PatchCore(FPFH+Raw) | 0.848 | 0.777 | 0.626 | 0.853 | 0.784 | 0.628 | 0.837 | 0.359 | 0.767 | 0.663 | 0.471 | 0.570 | 0.682 |
| PatchCore(PointMAE) | 0.726 | 0.498 | 0.585 | 0.827 | 0.783 | 0.489 | 0.630 | 0.374 | 0.539 | 0.501 | 0.519 | 0.663 | 0.594 |
| CPMF | 0.632 | 0.518 | 0.718 | 0.640 | 0.640 | 0.554 | 0.840 | 0.349 | 0.843 | 0.393 | 0.526 | 0.845 | 0.625 |
| IMRNet | 0.762 | 0.711 | 0.755 | 0.780 | 0.905 | 0.517 | 0.880 | 0.674 | 0.604 | 0.665 | 0.674 | 0.774 | 0.725 |
| **Reg3D-AD** | 0.716 | 0.697 | 0.827 | 0.852 | 0.900 | 0.584 | 0.915 | 0.417 | 0.762 | 0.583 | 0.506 | 0.685 | 0.704 |
| **Group3AD(Ours)** | 0.744 | 0.728 | 0.847 | 0.786 | 0.932 | 0.679 | 0.976 | 0.539 | 0.841 | 0.585 | 0.562 | 0.796 | 0.751 |

**Table 2: O-AUPR score (↑) on Real3D-AD. The best and second-best results are marked in red and blue, respectively.**

| Method | Airplane | Car | Candybar | Chicken | Diamond | Duck | Fish | Gemstone | Seahorse | Shell | Starfish | Toffees | Average |
|---|---|---|---|---|---|---|---|---|---|---|---|---|---|
| BTF(Raw) | 0.506 | 0.523 | 0.490 | 0.464 | 0.535 | 0.760 | 0.633 | 0.598 | 0.793 | 0.751 | 0.579 | 0.700 | 0.611 |
| BTF(FPFH) | 0.659 | 0.653 | 0.638 | 0.814 | 0.677 | 0.620 | 0.638 | 0.603 | 0.567 | 0.434 | 0.557 | 0.505 | 0.614 |
| M3DM(PointBERT) | 0.497 | 0.517 | 0.480 | 0.716 | 0.661 | 0.569 | 0.628 | 0.628 | 0.491 | 0.638 | 0.573 | 0.569 | 0.581 |
| M3DM(PointMAE) | 0.479 | 0.508 | 0.498 | 0.739 | 0.620 | 0.533 | 0.525 | 0.663 | 0.518 | 0.616 | 0.573 | 0.593 | 0.572 |
| PatchCore(FPFH) | 0.852 | 0.611 | 0.553 | 0.872 | 0.569 | 0.506 | 0.642 | 0.411 | 0.508 | 0.573 | 0.491 | 0.506 | 0.591 |
| PatchCore(FPFH+Raw) | 0.807 | 0.766 | 0.611 | 0.885 | 0.767 | 0.560 | 0.844 | 0.411 | 0.763 | 0.553 | 0.473 | 0.559 | 0.667 |
| PatchCore(PointMAE) | 0.747 | 0.555 | 0.576 | 0.864 | 0.801 | 0.488 | 0.720 | 0.444 | 0.546 | 0.590 | 0.561 | 0.708 | 0.633 |
| **Reg3D-AD** | 0.703 | 0.753 | 0.824 | 0.884 | 0.884 | 0.588 | 0.939 | 0.454 | 0.787 | 0.646 | 0.491 | 0.721 | 0.723 |
| **Group3AD(Ours)** | 0.757 | 0.706 | 0.837 | 0.674 | 0.932 | 0.612 | 0.981 | 0.533 | 0.842 | 0.648 | 0.567 | 0.785 | 0.740 |

**Table 3: P-AUROC score (↑) on Real3D-AD. The best and second-best results are marked in red and blue, respectively.**

| Method | Airplane | Car | Candybar | Chicken | Diamond | Duck | Fish | Gemstone | Seahorse | Shell | Starfish | Toffees | Average |
|---|---|---|---|---|---|---|---|---|---|---|---|---|---|
| BTF(Raw) | 0.564 | 0.647 | 0.735 | 0.608 | 0.563 | 0.601 | 0.514 | 0.597 | 0.520 | 0.489 | 0.392 | 0.623 | 0.571 |
| BTF(FPFH) | 0.738 | 0.708 | 0.864 | 0.693 | 0.882 | 0.875 | 0.709 | 0.891 | 0.512 | 0.571 | 0.501 | 0.815 | 0.730 |
| M3DM(PointBERT) | 0.523 | 0.593 | 0.682 | 0.790 | 0.594 | 0.668 | 0.589 | 0.646 | 0.574 | 0.732 | 0.563 | 0.677 | 0.636 |
| M3DM(PointMAE) | 0.530 | 0.607 | 0.683 | 0.735 | 0.618 | 0.678 | 0.600 | 0.654 | 0.561 | 0.748 | 0.555 | 0.679 | 0.637 |
| PatchCore(FPFH) | 0.471 | 0.643 | 0.637 | 0.618 | 0.760 | 0.430 | 0.464 | 0.830 | 0.544 | 0.596 | 0.522 | 0.411 | 0.577 |
| PatchCore(FPFH+Raw) | 0.556 | 0.740 | 0.749 | 0.558 | 0.854 | 0.658 | 0.781 | 0.539 | 0.808 | 0.753 | 0.613 | 0.549 | 0.680 |
| PatchCore(PointMAE) | 0.579 | 0.610 | 0.635 | 0.683 | 0.776 | 0.439 | 0.714 | 0.514 | 0.660 | 0.725 | 0.641 | 0.727 | 0.642 |
| CPMF | 0.618 | 0.836 | 0.734 | 0.559 | 0.753 | 0.719 | 0.988 | 0.449 | 0.962 | 0.725 | 0.800 | 0.959 | 0.758 |
| **Reg3D-AD** | 0.631 | 0.718 | 0.724 | 0.676 | 0.835 | 0.503 | 0.826 | 0.545 | 0.817 | 0.811 | 0.617 | 0.759 | 0.705 |
| **Group3AD(Ours)** | 0.636 | 0.745 | 0.738 | 0.759 | 0.862 | 0.631 | 0.836 | 0.564 | 0.827 | 0.798 | 0.625 | 0.803 | 0.735 |

**Table 4: P-AUPR score (↑) on Real3D-AD. The best and second-best results are marked in red and blue, respectively.**

| Method | Airplane | Car | Candybar | Chicken | Diamond | Duck | Fish | Gemstone | Seahorse | Shell | Starfish | Toffees | Average |
|---|---|---|---|---|---|---|---|---|---|---|---|---|---|
| BTF(Raw) | 0.012 | 0.014 | 0.025 | 0.049 | 0.032 | 0.020 | 0.017 | 0.014 | 0.031 | 0.011 | 0.017 | 0.016 | 0.022 |
| BTF(FPFH) | 0.027 | 0.028 | 0.118 | 0.044 | 0.239 | 0.068 | 0.036 | 0.075 | 0.027 | 0.018 | 0.034 | 0.055 | 0.064 |
| M3DM(PointBERT) | 0.007 | 0.017 | 0.016 | 0.377 | 0.038 | 0.011 | 0.039 | 0.017 | 0.028 | 0.021 | 0.040 | 0.018 | 0.052 |
| M3DM(PointMAE) | 0.007 | 0.018 | 0.016 | 0.310 | 0.033 | 0.011 | 0.025 | 0.018 | 0.030 | 0.022 | 0.040 | 0.021 | 0.046 |
| PatchCore(FPFH) | 0.027 | 0.034 | 0.142 | 0.040 | 0.273 | 0.055 | 0.052 | 0.093 | 0.031 | 0.031 | 0.037 | 0.040 | 0.071 |
| PatchCore(FPFH+Raw) | 0.016 | 0.160 | 0.092 | 0.045 | 0.363 | 0.034 | 0.266 | 0.066 | 0.291 | 0.049 | 0.035 | 0.055 | 0.123 |
| PatchCore(PointMAE) | 0.016 | 0.069 | 0.020 | 0.052 | 0.107 | 0.008 | 0.201 | 0.008 | 0.071 | 0.043 | 0.046 | 0.055 | 0.058 |
| CPMF | 0.010 | 0.064 | 0.050 | 0.031 | 0.074 | 0.018 | 0.559 | 0.007 | 0.636 | 0.025 | 0.128 | 0.391 | 0.166 |
| **Reg3D-AD** | 0.017 | 0.135 | 0.109 | 0.044 | 0.191 | 0.010 | 0.437 | 0.016 | 0.182 | 0.065 | 0.039 | 0.067 | 0.109 |
| **Group3AD(Ours)** | 0.018 | 0.174 | 0.122 | 0.068 | 0.287 | 0.016 | 0.448 | 0.009 | 0.24 | 0.067 | 0.056 | 0.134 | 0.137 |

*4.2.2 Evaluating Intercluster Uniformity & Intracluster Alignment Method for Anomaly Detection.* Our experimental analysis prominently features the evaluation of the Intercluster Uniformity Network (IUN) and Intracluster Alignment Network (IAN) through a series of ablation studies. By comparing the performance metrics of models with the IUN&IAN component both enabled and disabled, From Table 5, we observed a significant enhancement in the model's ability to discriminate between normal and anomalous features when the IUN&IAN was active. This enhancement directly correlates with IUN&IAN's primary function: to optimize the separation and uniform distribution of feature clusters in the

**Table 5: Ablation studies on the INU & IAN Methods. The best and second-best results are marked in red and blue, respectively.**

| Metric | Method | Airplane | Car | Candybar | Chicken | Diamond | Duck | Fish | Gemstone | Seahorse | Shell | Starfish | Toffees | Average |
|---|---|---|---|---|---|---|---|---|---|---|---|---|---|---|
| O-AUROC (↑) | PatchCore(PointMAE) | 0.726 | 0.498 | 0.585 | 0.827 | 0.783 | 0.489 | 0.630 | 0.374 | 0.539 | 0.501 | 0.519 | 0.663 | 0.594 |
| | PatchCore(PointMAE) with IUN&IAN | 0.748 | 0.558 | 0.656 | 0.655 | 0.845 | 0.579 | 0.741 | 0.478 | 0.556 | 0.522 | 0.53 | 0.662 | 0.628 |
| | Group3AD without IUN&IAN | 0.721 | 0.712 | 0.842 | 0.773 | 0.897 | 0.656 | 0.965 | 0.511 | 0.838 | 0.536 | 0.553 | 0.775 | 0.732 |
| | **Group3AD** | 0.744 | 0.728 | 0.847 | 0.786 | 0.932 | 0.679 | 0.976 | 0.539 | 0.841 | 0.585 | 0.562 | 0.796 | 0.751 |
| P-AUPR (↑) | PatchCore(PointMAE) | 0.016 | 0.069 | 0.020 | 0.052 | 0.107 | 0.008 | 0.201 | 0.008 | 0.071 | 0.043 | 0.046 | 0.055 | 0.058 |
| | PatchCore(PointMAE) with IUN&IAN | 0.013 | 0.066 | 0.021 | 0.047 | 0.118 | 0.010 | 0.210 | 0.009 | 0.081 | 0.037 | 0.063 | 0.12 | 0.066 |
| | Group3AD without IUN&IAN | 0.011 | 0.158 | 0.100 | 0.051 | 0.210 | 0.014 | 0.420 | 0.010 | 0.240 | 0.037 | 0.033 | 0.101 | 0.115 |
| | **Group3AD** | 0.018 | 0.174 | 0.122 | 0.068 | 0.287 | 0.016 | 0.448 | 0.009 | 0.240 | 0.067 | 0.056 | 0.134 | 0.137 |

**Table 6: Ablation studies on the AGCS scheme. The best and second-best results are marked in red and blue, respectively.**

| Metric | Method | Airplane | Car | Candybar | Chicken | Diamond | Duck | Fish | Gemstone | Seahorse | Shell | Starfish | Toffees | Average |
|---|---|---|---|---|---|---|---|---|---|---|---|---|---|---|
| O-AUROC (↑) | PatchCore(PointMAE) | 0.726 | 0.498 | 0.585 | 0.827 | 0.783 | 0.489 | 0.630 | 0.374 | 0.539 | 0.501 | 0.519 | 0.663 | 0.594 |
| | PatchCore(PointMAE) with AGCS | 0.713 | 0.544 | 0.614 | 0.676 | 0.820 | 0.544 | 0.752 | 0.452 | 0.590 | 0.507 | 0.524 | 0.778 | 0.626 |
| | Group3AD without AGCS | 0.723 | 0.708 | 0.833 | 0.726 | 0.906 | 0.661 | 0.941 | 0.523 | 0.818 | 0.544 | 0.531 | 0.787 | 0.725 |
| | **Group3AD** | 0.744 | 0.728 | 0.847 | 0.786 | 0.932 | 0.679 | 0.976 | 0.539 | 0.841 | 0.585 | 0.562 | 0.796 | 0.751 |
| P-AUPR (↑) | PatchCore(PointMAE) | 0.016 | 0.069 | 0.020 | 0.052 | 0.107 | 0.008 | 0.201 | 0.008 | 0.071 | 0.043 | 0.046 | 0.055 | 0.058 |
| | PatchCore(PointMAE) with AGCS | 0.010 | 0.057 | 0.022 | 0.055 | 0.145 | 0.013 | 0.261 | 0.009 | 0.092 | 0.040 | 0.039 | 0.137 | 0.073 |
| | Group3AD without AGCS | 0.019 | 0.176 | 0.067 | 0.058 | 0.253 | 0.015 | 0.437 | 0.008 | 0.237 | 0.048 | 0.042 | 0.124 | 0.124 |
| | **Group3AD** | 0.018 | 0.174 | 0.122 | 0.068 | 0.287 | 0.016 | 0.448 | 0.009 | 0.240 | 0.067 | 0.056 | 0.134 | 0.137 |

feature space. While the IUN&IAN establishes a foundational separation and distribution of feature clusters, the IAN gently fine-tunes this landscape, promoting tighter and more cohesive clusters. Such optimization evidently aids in reducing ambiguity and overlap between clusters, thereby sharpening the model's anomaly detection capabilities. The empirical evidence from our ablation studies thus underscores the essential role of IUN&IAN in fortifying the model's performance, highlighting its value in the context of HR 3D-AD.

*4.2.3 Evaluating Adaptive Group-Center Selection for Anomaly Detection.* In our comprehensive analysis, we further examined the role of the Adaptive Group-Center Selection (AGCS) within our proposed Group3AD. The AGCS enhances the detection process by strategically directing the model's focus towards regions potentially harboring anomalies, thus adjusting the sampling density in critical areas. Experimental results shown in Table 6 indicate that AGCS substantially improves the model's effectiveness in detecting anomalies. By concentrating on areas with potential irregularities, AGCS allows the model to identify subtle yet critical anomalies that might be missed under uniform sampling conditions. This prioritization leads to a more efficient detection process, enabling the model to perform more accurately and swiftly, especially in complex point cloud datasets.

## 4.3 Visualization

Figure 5 provides a visual representation of the clustering results achieved with the Group3AD framework on the Real3D-AD dataset, focusing on the effectiveness of the Intercluster Uniformity Network (IUN) and Intracluster Alignment Network (IAN). Each dot signifies a group center, pinpointed through the refined feature space sculpted by the IUN and IAN. This visualization underscores the success of these networks in creating distinct, well-separated clusters while maintaining tight intra-cluster relationships, which are instrumental for the precise localization of anomalies. The clear separation of clusters showcases the IUN's impact in enhancing feature discrimination across groups, while the density of points within each cluster attests to the IAN's role in fostering coherence

among features, resulting in improved anomaly detection performance in high-resolution 3D contexts.

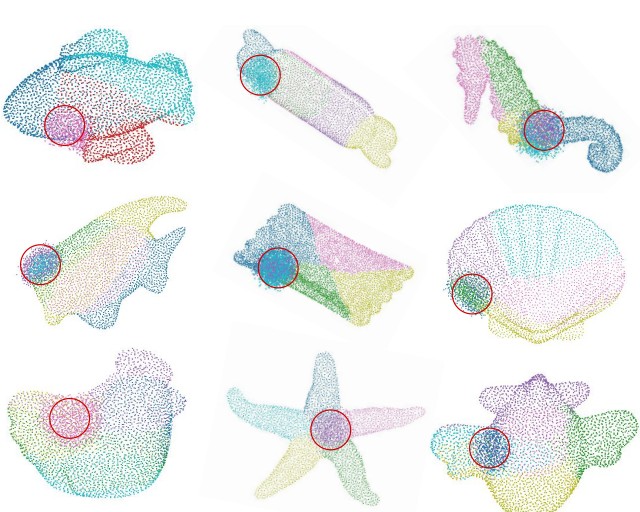

**Figure 5: Visualization results obtained by Group3AD. Different colors indicate different groups selected by AGCS. The red circle represents anomaly area.**

## 5 CONCLUSIONS

In this work, we introduced Group3AD, a robust framework tailored for high-resolution 3D-AD in industry, which can effectively enhance anomaly detection and localization accuracy. Demonstrating significant improvements over existing methodologies, especially in terms of reducing false positives and leveraging depth information for clearer anomaly identification, Group3AD emerges as a practical solution for real-world applications. Future endeavors may extend its applicability and efficiency, further solidifying its utility in industrial anomaly detection tasks.

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
