# OpenReview forum: "Towards High-resolution 3D Anomaly Detection via Group-Level Feature Contrastive Learning"
_acmmm.org/ACMMM/2024/Conference — MM2024 Poster_

### Official Review · Reviewer_NJNB · 2024-05-02

**Rating:** 4
**Confidence:** 4

**Summary:**

Towards High-resolution 3D Anomaly Detection, this paper propose a novel group-level feature-based network, called Group3AD. The Intercluster Uniformity Network is proposed to separate different groups and then obtain a more uniform distribution between clusters. The Intraclsuter Alignment Network is proposed to encourage groups within the cluster to be distributed tightly. The Adaptive Group-Center Selection can base on geometric information to improve the pixel density of potential anomalous regions during inference.

**Strengths:**

The propose method can achieve good results on the Real3D-AD dataset. The IUN, IAN, and AGCS modules are plugin-in can also be used in other methods. The ablation studies show that these modules are also effective when used in the PatchCore.

**Limitations:**

[Major]
1. As a 3D anomaly detection method, why is there no experiment results on the MVTec-3D dataset? In Table 2, why are the results of CPMF and IMRNet are missing? In Table 3 and 4, why are the results of IMRNet are missing? Why is the P-AUROC of Group3AD significantly higher than that of CPMF, but P-AUROC and P-AUPR is much lower?

2. During inference, after obtaining the center points, how to compute the anomaly scores based on the center points? These details should be described in the paper. Or, if the anomaly scoring is the same as in other 3D AD methods, you should note this in the paper.

3. Implementation details are missing. How to set the attention factor $\alpha$ in AGCS, ablation studies should be conducted. I think that the method in this paper is hard to reproduce only based on the descriptions in the main text, so will the code be open source?

4. Why do we need to produce pseudo anomalies during training? Is it beneficial for feature separation or something else? Ablation studies are required.

5. In line 692-693, the improvements in the Tables are 4.7%, 1.7%, 3%, 2.8%, but you stated as 6.7%, 2.4%, 4.3%, 25.7%. The improvements should not calculated by dividing the baselines, it’s confusing and also exaggerated.

[Minor]

1.The Figure 2 is not very clear to understand. Why is there a Coordinate Feature Extraction branch in the top, what is the difference with the Group-level Feature Extraction branch. The points in the memory bank also seems very messy, making it hard to see what the intention is. It would be better to add more explanations in the caption or modify this figure to make it clearer to understand.

My main concerns are in the [Major] part, the [Minor] part contains some suggestions, it would be ok to not respond the [Minor] part.

**Suitability:**

2

---

### Official Review · Reviewer_7NR1 · 2024-05-23

**Rating:** 3
**Confidence:** 3

**Summary:**

High-resolution point clouds anomaly detection plays a critical role in precision machining and high-end equipment manufacturing. The challenges of large amounts of points at the sample level, degradation of the representation, and small abnormal areas cause the difficulty for the detection. The paper proposes a novel group-level feature-based network to address the HRPCD-AD task. The Intercluster Uniformity Network, Intracluster Alignment Network, and Adaptive Group-Center Selection are proposed to improve the detection performance. However, the innovation is limited, and the only one dataset is used to evaluate the algorithm performance, maybe the more datasets should be verified for the proposed algorithm.

If the robustness of the proposed algorithm needs to be proven, more datasets should be used for validation. A single dataset cannot demonstrate the robustness and stability of the approach.

In comparative experiments, more classic and state-of-the-art algorithms also need to be compared. This is necessary to comprehensively demonstrate the advancement of the proposed algorithm. Therefore, it is recommended to further supplement the comparative experiments to make the advancement of the proposed algorithm more convincing.

In comparative experiments, adding instances of different algorithms for comparison will help to more clearly demonstrate the effectiveness of the proposed algorithm. If the overall effect is not very clear, is it possible to showcase it through local magnification? If this can be achieved, it would be a great bonus.

**Strengths:**

A novel group-level feature-based network is proposede to address the HRPCD-AD task, and the effectiveness of the proposed Group3AD surpasses Reg3D-AD by the margin of 5% in terms of object-level AUROC on Real3D-AD.

**Limitations:**

The proposed algorithm should be verified by the more datasets.

**Suitability:**

2

---

### Official Review · Reviewer_ZMdx · 2024-06-04

**Rating:** 4
**Confidence:** 3

**Summary:**

The paper proposes a method for 3D anomaly detection on high-resolutions point cloud, which is more likely related to practical industrial applications. By introducing group-level feature learning based on two designed losses for inter-class and intra-class feature widening and alignment, the encoder can extract features that are more distinctive for anomaly detection. During inference, an adaptive sampling strategy is also proposed to let the model focus on potential anomaly areas. Experiments show that the proposed method gives better performance than existing ones, on high-resolution 3D point cloud dataset.


Overall, this paper has good organizations, motivations, detailed methods, and experiments. I would like to see the responses from the authors.

**Strengths:**

The method is straightforward and intuitive. Using clusters and learn features to enlarge distance between clusters and enhance compactness within clusters is though not new in traditional machine learning methods, while adopting this mechanism to the high-resolution 3D anomaly detection task might be new. Experimental results also show its good performance.

**Limitations:**

1.	Same as stated in the pros, the cluster-based feature learning mechanism might not be a new thing. An old idea is applied in a new task, though widely seen in recent papers, might be limiting the paper’s novelty.

2.	It might be misleading to state that the AGCS enhances detection efficiency, as there are no experimental results show that. It is suggested to weaken this statement, or give more evidence to prove this (by additional discussions or extra experiments).

Minor question:

3.	If I understand correctly, are all the points used in the training process (if all the points in each cluster are considered when calculating the losses)?

**Suitability:**

2

---

### Official Review · Reviewer_UNzY · 2024-06-08

**Rating:** 4
**Confidence:** 1

**Summary:**

he paper presents a novel framework called Group3AD for high-resolution 3D anomaly detection (HRPCD-AD) task.​ The framework includes three key designs: Intercluster Uniformity Network (IUN) and Intracluster Alignment Network (IAN) to enhance the uniformity and alignment of feature distribution, and Adaptive Group-Center Selection (AGCS) to focus on regions with potential anomalies during inference.

**Strengths:**

- The group-level feature contrastive learning framework is novel.

- The Adaptive Group-Center Selection (AGCS) that prioritizes regions with potential anomalies can help improve the detection efficiency.
​

**Limitations:**

- Limited Discussion on Implementation Details: The paper does not provide detailed information about the implementation of the proposed framework, such as network architectures, training procedures, and hyperparameter settings. This makes it difficult for readers to replicate and reproduce the results.

**Suitability:**

2

---

### Meta-Review · Area_Chair_nAGM · 2024-06-25

**Recommendation:** Accept (Poster)
**Confidence:** 5

**Metareview:**

The paper proposes a novel approach, Group3AD, for high-resolution 3D anomaly detection, demonstrating significant improvements over existing methods. While some concerns regarding implementation details and robustness across multiple datasets were noted, the authors' rebuttal addressed these adequately. The method's contribution to the field justifies its acceptance.